# Metabolism, Transport and Drug–Drug Interactions of Silymarin

**DOI:** 10.3390/molecules24203693

**Published:** 2019-10-14

**Authors:** Ying Xie, Dingqi Zhang, Jin Zhang, Jialu Yuan

**Affiliations:** School of Pharmacy, Macau University of Science and Technology, Taipa, Macao 999078, China; zhangdingqi1998@gmail.com (D.Z.);

**Keywords:** silybin/silymarin, metabolism, efflux transporters, drug–drug interaction (DDI)

## Abstract

Silymarin, the extract of milk thistle, and its major active flavonolignan silybin, are common products widely used in the phytotherapy of liver diseases. They also have promising effects in protecting the pancreas, kidney, myocardium, and the central nervous system. However, inconsistent results are noted in the different clinical studies due to the low bioavailability of silymarin. Extensive studies were conducted to explore the metabolism and transport of silymarin/silybin as well as the impact of its consumption on the pharmacokinetics of other clinical drugs. Here, we aimed to summarize and highlight the current knowledge of the metabolism and transport of silymarin. It was concluded that the major efflux transporters of silybin are multidrug resistance-associated protein (MRP2) and breast cancer resistance protein (BCRP) based on results from the transporter-overexpressing cell lines and MRP2-deficient (TR^−^) rats. Nevertheless, compounds that inhibit the efflux transporters MRP2 and BCRP can enhance the absorption and activity of silybin. Although silymarin does inhibit certain drug-metabolizing enzymes and drug transporters, such effects are unlikely to manifest in clinical settings. Overall, silymarin is a safe and well-tolerated phytomedicine.

## 1. Introduction

Silymarin, an extraction from the seeds of milk thistle (*Silybum marianum*), has been used for liver and gallbladder dysfunction in Europe for thousands of years [1,2]. Silybin, also known as silibinin, is the major bioactive flavonolignan in silymarin, comprising approximately 50–70% of the extract [3]. Naturally-occurring silybin is a mixture of two diastereomers named silybin A and silybin B in approximately equal portions. They have configurations of 2*R*, 3*R*, 10*R*, 11*R* and 2*R*, 3*R*, 10*S*, 11*S*, respectively, as shown in Figure 1 [4]. Other flavonolignans such as isosilybin A and B, silychristin, isosilychristin, silydianin, and silimonin are also present in silymarin [5].

In recent years, studies have demonstrated that the pharmacological effects of silymarin/silybin are not limited to the treatment of liver diseases. It also has equally promising effects in protecting the pancreas, kidney, myocardium, and central nervous system [6]. This is because silymarin/silybin has antioxidant, anti-inflammatory, protein synthesis-enhancing, and anti-fibrotic activities as well as a good safety profile [7,8,9]. Moreover, silymarin/silybin can inhibit gluconeogenesis and autophagy of pancreatic β-cells, and repair damaged cells, showing anti-diabetic effects [10,11]. Recently, studies further demonstrated the anti-cancer effects of silybin via regulating the cancer cell cycle, apoptosis and autophagy, as well as inhibiting tumor-inducing factors [12,13,14]. Moreover, it is interesting to note that silybin has potential therapeutic effects for Alzheimer’s disease (AD) via counteracting the toxicity of Aβ (amyloid beta) [15] which is the central peptide responsible for AD [16].

Although preclinical pharmacological benefits of silymarin seem promising, few of them have manifested in the clinical studies. Higher-than-usual-dose silymarin failed to produce a satisfactory anti- hepatitis C virus (HCV) effect in a large randomized controlled trial [17]. This is possibly because of the poor bioavailability of silymarin, which led to plasma concentrations far below the levels used in in vitro experiments [18,19,20]. After absorption, silymarin undergoes rapid phase II metabolism and is primarily excreted into bile and urine. Moreover, it exhibits enhanced absorption in patients with hepatitis C and nonalcoholic fatty liver disease [21,22]. Therefore, understanding the metabolism and transport of silymarin will be important for improving its low bioavailability and resolving the inconsistencies in clinical outcomes as well as results of in vitro studies. Moreover, it is necessary to understand the risks of drug–drug interactions (DDIs) associated with the ingestion of silymarin/silybin, if there are any, in which a dose adjustment may be required.

The goal of the present paper is to summarize and highlight the current knowledge of the metabolism and transport of silymarin/silybin and drug–drug interactions mediated by metabolizing enzymes and transporters.

## 2. Metabolism and Transport of Silymarin

Silymarin has oral absorption only about 23–47% and quick phase II conjugation, leading to low bioavailability, which has been reviewed [20]. Here, we would like to provide a more dedicated and up-to-date outlook on the metabolism and transport of silymarin.

### 2.1. Absorption

It is commonly agreed that silymarin suffers from low bioavailability due to poor solubility in water [1]. However, efflux transporters on the apical side of the intestinal epithelium further throttled the absorption of silybin as seen in Figure 2 [19,23].

The Caco-2 cell monolayer model was used to investigate the intestinal absorption of silybin [23]. The measured mean efflux ratios of silybin A and silybin B were 5.05 and 4.61, respectively, indicating an active transport mechanism. Moreover, MK571 (a multidrug resistance-associated protein (MRP2)-specific inhibitor) significantly decreased the efflux ratio of silybin, while Ko143 (a breast cancer resistance protein (BCRP)-specific inhibitor) and cyclosporin A (both a BCRP and P-glycoprotein (P-gp) inhibitor) were less potent, suggesting that intestinal efflux of silybin is mediated by MRPs and possibly BCRP. Studies carried out with Madin-Darby canine kidney cells II (MDCKII) lines overexpressing transporters (MRP2, BCRP, or MDR1) and a sandwich-cultured hepatocyte model confirmed that the transporters involved in the absorption and excretion of silybin are MRP2 and BCRP, but not P-gp [23], raising the question of whether inhibitors of these two transporters are likely to increase silybin absorption.

### 2.2. Metabolism

After oral administration, silybin/silymarin undergoes both phase I and phase II biotransformation, especially the latter [24]. Figure 3 shows that phase I metabolites of silybin mainly include O-demethylated ones mediated by the CYP2C8 (Cytochrome P450 Family 2 Subfamily C Member 8) isoenzyme [25]. In addition, four minor metabolites including three monohydroxy ones and one dihydroxy one are also observed, though their structures are not confirmed [26]. Silybin and its phase I metabolites undergo extensive phase II biotransformation, as most of the silybin in the system exists as conjugates including 55% glucuronidated conjugates and about 28% sulfated ones [17,18].

Glucuronidation reactions of silybin are mediated by UDP-glucuronosyltransferase (UGT)1A1, 1A6, 1A7, 1A9, 2B7, and 2B15, while sulfidation reactions are mediated by sulfotransferases (SULTs) [27]. Four monoglucuronides, 1 diglucuronide, 3 monosulfates, 2 glucuronide sulfates, and *O*-demethylated glucuronide have been detected in plasma after silymarin administration [28,29]. There are two major glucuronidation sites, C-7 and C-20. Silybin is glucuronidated in a stereoselective manner, with silybin B more efficiently glucuronidated at the C-20 position, while silybin A is glucuronidated similarly on both sites [4] due to the stereoselective activity of certain UGT isoforms [27].

The rapid and extensive phase II metabolism of silybin has been considered as the major reason contributing to its low bioavailability. However, in a group of 56 subjects diagnosed with either HCV or non-alcoholic fatty liver disease (NAFLD), mutant allele UGT1A1*28 produced little inter-subject variability in silybin pharmacokinetics [30]. These data indicate that the role of metabolic enzyme activity is obscured by other more prevalent factors.

### 2.3. Elimination

Both free and conjugated silymarin eliminated rapidly in vivo. However, the renal excretion of silybin is low and accounts for only 1–2% of an original oral dose over 24 h [31]. Instead, hepatobiliary excretion is extensive, as biliary concentration of silybin was about 100 times higher than serum concentration in patients [31]. Based on an AUC_bile_/AUC_blood_ (blood-to-bile distribution ratio) distribution ratio (30 ± 9.4) of total silybin, silybin was proposed to be excreted into the bile through active transport [32]. As with most flavonoids, the pharmacokinetic behavior of silybin in vivo exhibits a enterohepatic circulation, where excreted glucuronidated silybin undergoes bacterial enzymatic cleavage of β-glucosidic bonds and is re-absorbed, as indicated by the secondary peak in the plasma concentration curve [21].

Miranda et al. showed that MRP2 was the transporter mainly responsible for the biliary excretion of free and conjugated silybin using perfused rat liver from wild type and MRP2-deficient (TR^−^) Wistar rats [33]. Whether or to what extent BCRP is involved in biliary excretion of silybin remains unknown [23]. By targeting efflux transporters responsible for the excretion and absorption of silybin and its metabolites, researchers demonstrated that bioavailability enhancers, or bio-enhancers, boost bioavailability and pharmacological effects of silybin in vivo.

## 3. Silybin as a Beneficiary of DDI

Previous studies focus extensively on silybin interfering with the metabolism and disposition of other drugs. However, some studies in recent years revealed the likelihood that silybin may be influenced by certain compounds such as tangeretin [23], piperine [34], and baicalein [19]. New formulations of silybin containing these bio-enhancers [35] provide new therapeutic strategy.

### 3.1. Tangeretin

Tangeretin is a flavonoid that mainly exists in citrus fruits, especially in the peel [36]. Yuan et al. found that co-administration of tangeretin not only significantly increased absorption and bioavailability of silybin, but also enhanced the hepatoprotective, anti-inflammatory, and antioxidant effect of silybin in a CCl_4_-induced liver injury rat model. Tangeretin could enhance the intestinal absorption and inhibit the biliary excretion of both free and conjugated silybin based on the data from Caco-2 and sandwich-cultured rat hepatocyte (SCH) models by inhibiting the efflux transporters MRPs and BCRP [23].

### 3.2. Piperine

Piperine is the main component of black and long pepper that has attracted much attention in recent years for its inhibiting activities on efflux transporters and metabolic enzymes [37]. Bi and colleagues proved that piperine is not only an inhibitor of P-gp, but also an inhibitor of MRP2 and BCRP in Caco-2 model and MDCKII cell lines [34]. When silybin and piperine were co-administered, the maximum plasma drug concentration (C_max_) and the area under the drug concentration–time curve up to time ‘t’( AUC_0-t_) of both silybin A and silybin B in rats were higher than that of silybin used alone [34]. A 76% increase in C_max_ and a 37% increase in AUC_0-t_ of silybin A were observed, while C_max_ and AUC_0-t_ of total silybin B were increased by 60% and 26%, respectively. The concentrations of free silybin A and silybin B were also increased by piperine on a similar scale. MRP2 and BCRP are located on the apical membrane of intestinal epithelial cells and pumped substrates back into the intestine. Therefore, inhibition of MRP2 and BCRP by piperine may be a major reason for the increased exposure of silybin.

### 3.3. Baicalein and Baicalin

Baicalein (5,6,7-trihydroxy-2-phenyl-4H-1-benzopyran-4-one), which mainly exists in the genus *Scutellaria*, is a flavone with antioxidant activity and other pharmacological activities [38,39]. After oral administration, baicalein is glucuronidated into baicalin and baicalein 6-*O*-glucuronic acid by the UGT enzyme in either the intestine or the liver [40]. Both baicalein and baicalin are inhibitors of MRP2 and BCRP. Experiments have proved that when silybin and baicalein were administered simultaneously, the C_max_, AUC_0-t_, and area under the concentration time-curves from time zero to infinity (AUC_0-∞_) of silybin in rats were increased [19]. However, there might be other factors contributing to this result besides efflux transporter inhibition. Silybin and baicalein may compete for the binding site of the UGT enzyme in vivo [41], resulting in the increase of silybin concentration, which need further studies to confirm. The co-administration of baicalein and silybin showed a synergistic liver protective effects [19], suggesting a potential method for improving the clinical efficacy of silybin.

## 4. Silybin as a Perpetrator of DDI

Modulatory effects on the activity and expression of a wide spectrum of liver microsomal enzymes and efflux transporters were noticed with silybin or silymarin in the literature [42,43]. However, there is still a question as to whether silybin or silymarin have the potential to affect other clinical therapeutic drugs.

### 4.1. CYP-Mediated Drug–Drug Interaction

In vitro incubation revealed weak to moderate inhibition of silybin on most CYP enzymes. Silymarin or silybin inhibit CYP1A2, 2B6, 2C8, 2C9, 2C19, 2D6, and 3A4, but the most prominent inhibition effects are of CYP3A4 and CYP2C9 [42]. Depending on the study, the half maximal inhibitory concentration (IC_50_) for CYP2C9 was reported to be 34 μM or 43–45 μM ([25,42], respectively), and the IC_50_ for CYP 3A4 was reported to be 27 μM, 49.8 μM, or >200 μM ([25,42,44], respectively). However, contradictory inhibition results by silybin were observed for the two CYP3A4 substrates denitronifedipine (29 μM) and erythromycin (>200 μM), which may due to the existence of more than one binding site at CYP3A4 [42]. In general, it is commonly agreed that silybin suppresses the activity of most enzymes, but rarely at a clinically relevant concentration, which is up to 1.5 μM [45]. However, one exception, CYP2C9, was found vulnerable to silybin interference at a clinically relevant concentration (IC_50_ of 8.2 μM for silybin A), as was claimed in one study with warfarin in human liver microsomes [46]. It still needs confirmation based on data from the clinical study.

As for moderation of CYPs on a transcriptional level, downregulation of CYP 3A4 was only reported in Caco-2 monolayer [47], but not in primary human hepatocytes [48].

Numerous phase I clinical trials were performed to investigate silymarin/silybin-induced drug-drug interaction mainly based on CYP enzymes (Table 1). Aminopyrine and phenylbutazone used as non-specific probes were not affected by Legalon^®^ (silymarin) at 3 × 70 mg/day in a clinical study [49]. For CYP 3A4-mediated interactions, indinavir [50,51], midazolam [52,53], irinotecan [54], and ranitidine [55] were used as probe drugs. However, there are limited influences on the pharmacokinetics of these drugs, indicating no interaction of silymarin based on CYP 3A4. Similar conclusions were drawn for CYP 2D6 [52,56] and CYP 1A2 [52].

Despite clinical trials disproving interactions, animal experiments kept raising alarm. In rabbits, pretreatment but not co-administration of 50 mg/kg oral silybin increased nitrendipine (mostly metabolized by CYP3A4), area under the plasma drug concentration-t curve (AUC), and C_max_ [67]. A more recent study in rats observed an elevated exposure of the central analgesic methadone (mostly metabolized by CYP3A4) associated with pre-treatment of silybin [68].

In conclusion, silybin inhibits CYP enzymes, with 2C9 and 3A4 being the most sensitive, but rarely at clinically achievable concentrations. Thus, CYP-mediated DDI for silybin is not a major clinical concern, though sensitive drugs such as warfarin, opioids, and anti-arrhythmic agents require attention.

### 4.2. UGT-Mediated Drug–Drug Interaction

In comparison with CYPs, UGTs are generally more vulnerable to inhibition in the presence of silybin in in vitro environment. It was confirmed in Raman’s study that UGT 1A6/9 activity was reduced after silymarin treatment at a relatively high concentration of 0.1 mM [69]. However, Sirdar discovered that silybin is a potent inhibitor of recombinant UGT1A1, with an IC_50_ of only 1.4 μM, which is more selective than UGT 1A6/9 [70]. However, the clinical significance for inhibiting UGT1A1 was not clear.

Among these drug–drug interaction clinical trials associated with concomitant dosing of silybin or silymarin, however, few were carried out with UGT substrates (Table 1). There is only one study in which a dose of 3 × 200 mg/day milk thistle preparation (containing 80% silymarin) was given to six cancer patients receiving irinotecan therapy for 14 consecutive days [54]. No influence was observed on the pharmacokinetics of irinotecan after either short-term or long-term exposure of silybin in the form of milk thistle. However, the current clinical evidence cannot warrant that silybin has no interaction with UGT-metabolized drugs based on the limited data.

### 4.3. DDI Mediated by Transporters

Köch et al. discovered of the potent inhibition to the organic anion-transporting polypeptides (OATP) transporter family including OATP1B1, OATP1B3, and OATP2B1 by silymarin. However, the IC_50_ values of silymarin or silybin for OATPs were far below their portal vein concentrations in vivo, indicating that silymarin–drug interactions via inhibition of OATPs are unlikely to happen in clinical conditions with a customary dose [71].

Nguyen’s study showed that silybin reduced the efflux of two substrates of P-gp including digoxin and vinblastine in Panc-1 cells, indicating the inhibiting effect of silybin to P-gp [72]. However, most of the studies about P-gp inhibition by silybin or silymarin are pre-clinical studies with cells or animals. In fact, the results of the related clinical studies were complex. Gurley’s group revealed that milk thistle (containing 80% silymarin) at a dose given daily of 900 mg does not affect P-gp substrate digoxin [59] in terms of clinical related physiological parameters. However, Han’s group showed that silymarin significantly increased the plasma concentration of talinolol, which is also a typical substrate of P-gp, in healthy volunteers [62]. In contrast, Rajnarayana’s group found that silymarin increased the clearance of metronidazole with a concomitant decrease in half-life, C_max_, and AUC, as silymarin might induce both intestinal P-gp and CYP3A4 upon multiple-dose administration [58]. These variations may be due to many factors such as formulation, dose, frequency of administration, statistical methods, and so on.

## 5. Conclusions and Outlook

It is commonly viewed that silybin is well tolerated with few side effects, and has significant anti-inflammatory, antiviral, antioxidant, and anticancer activities. Silymarin, silybin, and milk thistle extract currently rank among the top-selling botanical supplements or phytomedicines. However, the clinical outcomes of the hepatoprotection effects are varied for silymarin, mainly caused by its low bioavailability. As a hepatoprotective agent, it is frequently prescribed as an adjuvant therapy along with other medications such as methotrexate, which has significant liver toxicity [73]. Therefore, it is necessary to understand the metabolism and/or transport of silymarin as well as the risk of DDI associated with the process.

Studies have shown that drugs inhibiting BCRP and MRP2, such as piperine or tangeretin [23,34], may lead to increased bioavailability of silybin. Similarly, enhanced-exposure of silybin occurred in patients with hepatic cirrhosis where transporter expression is also known to decrease [74]. However, mutant allele UGT1A1*28 produced little inter-subject variability in silybin pharmacokinetics [30]. Such increased silybin exposure is a result of silybin accumulation via efflux transport inhibition because of drug–drug interaction but is not related to the UGT enzyme.

From another point of view, our current findings on drug–drug interactions of silymarin, especially results from clinical trials, concluded that silymarin does not pose a clinically relevant risk of drug–drug interaction. Although silymarin does inhibit activities of enzymes and transporters concerned with the pharmacokinetics of therapeutic drugs, its concentration within the human body rarely reaches the point that constitutes significant inhibition due to low bioavailability of silymarin [45].

However, exceptions do exist. Pharmacokinetic studies carried out in healthy volunteers indicated that silymarin pretreatment increased the exposure of talinolol [62] and domperidone [65], but decreased the AUC of indinavir [57] and metronidazole [58]. In another study conducted with losartan, it was claimed that losartan biotransformation into its active form was significantly decreased by silybin administration, with the CYP2C9*1/*1 genotype more affected than the 2C9*1/*3 genotype [63]. Silybin also affects the pharmacokinetics of pyrazinamide, a chemotherapeutic agent for tuberculosis in rats [75]. These results remind us that the likelihood of DDI induced by silymarin/silybin persists, especially with certain sensitive drugs.

Since inconsistent results have been noted in different clinical studies regarding risks of DDI with medications such indinavir, further clinical studies are needed to investigate the impacts of pharmacogenetic factors, the pathological state of patients, and transporters. In addition, guidelines should be established for identifying drugs with potential silymarin-induced DDIs.

## Figures and Tables

**Figure 1 molecules-24-03693-f001:**
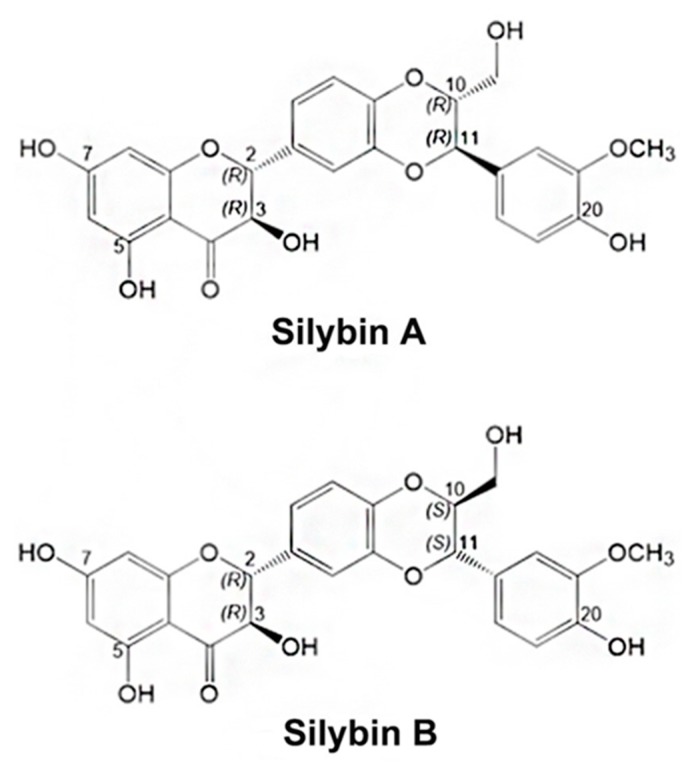
Chemical structures of silybin diastereomers.

**Figure 2 molecules-24-03693-f002:**
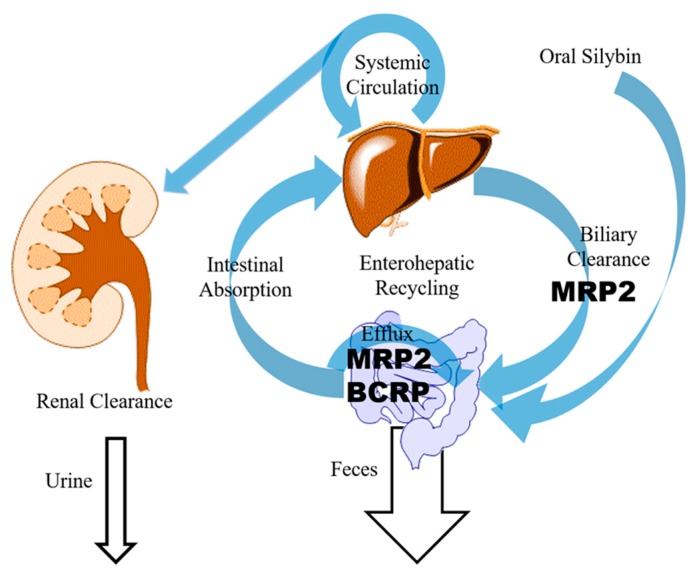
Transporters related to the disposition and elimination of silybin. BCRP: breast cancer resistance protein; MRP2: multidrug resistance-associated protein.

**Figure 3 molecules-24-03693-f003:**
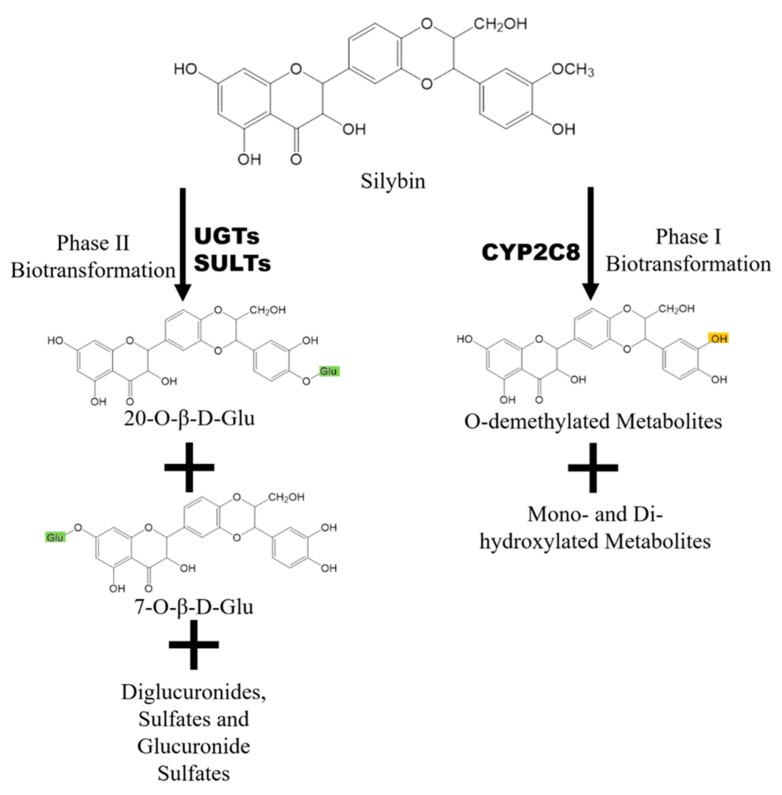
Metabolism of silybin and its major metabolites. UGT: UDP-glucuronosyltransferase; SULTs: sulfotransferases; CYP2C8: Cytochrome P450 2C8.

**Table 1 molecules-24-03693-t001:** List of published clinical trials on silybin-related drug–drug interactions *.

	Subjects	Silybin Dosing	Probe Drug Dosing	Enzymes or Transporters Involved	Conclusion	
1	16 healthy volunteers	3 × 70 mg Legalon^®^ (silymarin), 28 days	Aminopyrine/phenylbutazone		No influence	[49]
2	10 healthy volunteers	175 mg milk thistle extract, 3 times daily for 3 weeks	indinavir 800 mg/8 h	CYP3A4	9% reduction in AUC of indinavir	[50]
3	10 healthy volunteers	160 mg silymarin, 3 times/day	indinavir 800 mg 3 times/day	CYP3A4	No influence	[57]
4	16 healthy volunteers	450 mg milk thistle extract daily	indinavir	CYP3A4	No influence	[51]
5	12 healthy volunteers	140 mg silymarin daily for 9 days	400 mg metronidazole trice daily for 3 days	P-gp, CYP3A4, CYP2C9	28% reduction in AUC of metronidazole	[58]
6	12 healthy volunteers	175 mg (containing 80% silymarin) twice daily	midazolam and caffeine, followed 24 h later by chlorzoxazone and debrisoquin	CYP1A2, CYP2D6, CYP2E1, CYP3A4	No influence	[52]
7	6 cancer patients	200 mg milk thistle (containing 80% silymarin), thrice a day, for 14 consecutive days	irinotecan once a week i.v. 125 mg/m^2^	CYP3A4, UGT1A1	No influence	[54]
8	16 healthy volunteers	900 mg milk thistle (containing 80% silymarin) for 14 days	digoxin 0.4 mg	P-gp	No influence	[59]
9	19 healthy volunteers	900 mg milk thistle (containing 80% silymarin) for 14 days	midazolam	CYP3A	No influence	[53]
10	16 young male volunteers	280 mg silymarin	10 mg nifedipine	CYP3A4	No influence	[60]
11	12 young male volunteers	140 mg silymarin thrice daily	150 mg ranitidine	CYP3A4, P-gp	No influence	[55]
12	8 healthy male volunteers	140 mg silymarin 4 times daily	10 mg rouvastatin	OATP1B1, BCRP	No influence	[61]
13	16 healthy volunteers	300 mg milk thistle extract (containing 80% silymarin) 3 times daily	5 mg debrisoquine	CYP2D6	No influence	[56]
14	18 healthy adult men	140 mg silymarin 3 times daily for 14 days	talinolol	P-gp	Silymarin increased (36%) AUC of talinolol	[62]
15	12 healthy adult men	140 mg silymarin 3 times daily	losartan	CYP2C9	Inhibition CYP2C9 in a genotype-dependent manner	[63]
16	15 HIV-infected patients	150 mg silymarin 3 times daily	darunavir-ritonavir (600/100 mg twice daily)	CYP3A4, P-gp	Silymarin slightly decreased (15%) the AUC of darunavir-ritonavir	[64]
17	8 healthy male volunteers	500 mg silymarin twice daily for 7 days	10 mg domperidon	CYP3A4, P-gp	Silymarin pretreatment increased AUC of domperidone by 5-fold.	[65]
18	9 healthy volunteers	175 mg Legalon^®^ (140 mg silymarin) thrice daily for 14 days	caffeine, tolbutamide, dextromethorphan, midazolam	CYP1A2, CYP2C9, CYP2D6, CYP3A4/5	No influence	[66]

* Notes: only papers with the keyword silybin or silymarin, milk thistle, *Silybum marianum*, and silibinin in their titles were searched, which yielded 82 results. Among the 82 items only 18 studies satisfied our criteria as clinical trials studying the effects of silybin or silymarin on the pharmacokinetics of other drugs, and were organized into this table that displays their dosing regimens, probe drugs, enzymes or transporters that they studied, and conclusions. We noticed that clinical research on silybin drug–drug interactions (DDIs) has stagnated since 2014. As for dosing of silybin, dosing regimens range from an equivalence of 140 mg of standardized silymarin daily to 900 mg daily. Study designs include both crossover studies and parallel designs, either open-labeled or blinded. AUC: area under the plasma drug concentration-t curve; BCRP: breast cancer resistance protein; P-gp: P-glycoprotein; OATP: organic-anion-transporting polypeptide, CYP: cytochrome P-450; i.v.: Intravenous perfusion.

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
