# Peer review of "Metabolism, Transport and Drug–Drug Interactions of Silymarin"

_molecules, 2019, doi:10.3390/molecules24203693_

Round 1
Reviewer 1 Report
This is an interesting manuscript where the latest research on silybin metabolism is reported. The manuscript is well-read and has a logical sound. It highlights the metabolism of this interesting natural molecule by highlighting all its applications as a natural drug.
The authors should report in the introduction of the latest findings on silybin. Recently in vivo and in vitro studies have shown the ability of silybin (doi:10.1021/acschemneuro.7b00110) to inhibit the toxicity of Abeta (1-49), the polypeptide responsible for Alzheimer's disease (doi:10.1016/j.bbamem.2018.02.022). I suggest inserting this finding after describing diabetes since these diseases are interacting. English should be improved.
Reviewer 2 Report
This review describes the fate of silymarin in human organism.
1. Introduction : it is not easy for the reader who discovers silymarin in this paper to understand that silymarin designates a mixture of flavanolignans. The word "silibinin" designates the 2 diastereomers silybin A and silybin B, whereas other flavanolignans such as isosilybin A and B, silydianin and silychristin are also present in silymarin. This could be clearly explained.
2. Page 5, line 121 (end of paragraph on tangeretin) : the reference 20 by Yuan et al (2018) may be cited.
3. Page 6, lines 152-159 (§ 4.1) : the concentrations of silybin should be always in the same unit; this would be easier to compare the concentrations.
4. Page 6,line 165 : Legalon® is a trade mark, so it should have a «®».
5. Page 6, lines 170-177 : do the authors have an explanation for the contradictory results regarding the interaction of silybin with the different CYPs in humans and animals ?
6. Page 7, line 214 : replace "exaction" by "extract".
7. Page 8, line 205 : spelling mistake : P-gp
8. English should be improved.
